# Rapid Sterilization of Clinical Apheresis Blood Products Using Ultra-High Dose Rate Radiation

**DOI:** 10.3390/ijms26062424

**Published:** 2025-03-07

**Authors:** Stavros Melemenidis, Khoa D. Nguyen, Rosella Baraceros-Pineda, Cherie K. Barclay, Joanne Bautista, Hubert D. Lau, M. Ramish Ashraf, Rakesh Manjappa, Suparna Dutt, Luis A. Soto, Nikita Katila, Brianna Lau, Vignesh Viswanathan, Amy S. Yu, Murat Surucu, Lawrie B. Skinner, Edgar G. Engleman, Billy W. Loo, Tho D. Pham

**Affiliations:** 1Department of Radiation Oncology, Stanford University School of Medicine, Stanford, CA 94305, USA; stavmel@stanford.edu (S.M.); ramish.ashraf@stanford.edu (M.R.A.); rakesh9920@gmail.com (R.M.); sdutt@stanford.edu (S.D.); luisasr@stanford.edu (L.A.S.); nikita3@stanford.edu (N.K.); brianna.lau@my.rfums.org (B.L.); vignesh1984@gmail.com (V.V.); amysyu@stanford.edu (A.S.Y.); surucu@stanford.edu (M.S.); lawrie.skinner@stanford.edu (L.B.S.); bwloo@stanford.edu (B.W.L.); 2Stanford Blood Center, Stanford Health Care, Stanford, CA 94304, USA; khoa@stanford.edu (K.D.N.); cbarclay@stanford.edu (C.K.B.); joannedr@stanford.edu (J.B.); edgareng@stanford.edu (E.G.E.); 3Department of Health Policy, Center for Innovation to Implementation, Veterans Affairs Health Care, Palo Alto, CA 94304, USA; rosella.baraceros-pineda@va.gov; 4Department of Pathology, Stanford University School of Medicine, Stanford, CA 94305, USA; hlau@stanford.edu; 5Stanford Cancer Institute, Stanford University School of Medicine, Stanford, CA 94305, USA; 6Division of Immunology & Rheumatology, Department of Internal Medicine, Stanford University School of Medicine, Stanford, CA 94305, USA

**Keywords:** ultra-high dose rate (UHDR), radiation-based sterilization, blood products, bacteria-spiked platelets, COVID-19 convalescent plasma (CCP)

## Abstract

Blood products, including apheresis platelets and plasma, are essential for medical use but pose risks of bacterial contamination and viral transmission. Platelets are prone to bacterial growth due to their storage conditions, while plasma requires extensive screening. This study explores rapid irradiation as an innovative pathogen reduction method. A clinical linear accelerator was configured to deliver ultra-high dose rate (6 kGy/min) irradiation to platelet and plasma components. Platelets spiked with *Escherichia coli* (*E. coli*; 10⁵ colony-forming units) were irradiated at 0.1–20 kGy, followed by bacterial growth and platelet count analysis. COVID-19 convalescent plasma (CCP) was irradiated at 25 kGy, and receptor-binding domain (RBD)-specific immunoglobulins (Ig) were assessed. Irradiation at 1 kGy reduced *E. coli* growth by 2.7-log without significant platelet loss, while 5 kGy achieved complete suppression. The estimated 6-log bacterial reduction dose (2.3 kGy) led to a 31% platelet count drop. Administering a 25 kGy virus-sterilizing dose to CCP resulted in a 9.2% decrease in RBD-specific IgG binding. This study demonstrates the proof-of-concept for rapid blood sterilization using a clinical linear accelerator. The method maintains platelet counts and CCP antibody binding at sterilizing doses, highlighting its potential as a point-of-care blood product sterilization solution.

## 1. Introduction

In the crucial domain of transfusion medicine, whole blood and its derivative products like red blood cells, platelets, and plasma stand as the cornerstones of life-saving interventions. However, there are also life-threatening risks associated with blood transfusions.

The rates of transfusion transmitted infections (TTIs) by known viruses such as human immunodeficiency virus (HIV), hepatitis B virus (HBV), and hepatitis C virus (HCV) have been reduced over the decades through a combination of infectious disease screening and donor history questionnaires [1,2,3]. However, despite these improvements, there remains a residual risk, which refers to the chance of transmitting a virus from donor blood that has passed screening undetected, typically because the donor is in the window period of infection when the virus is not yet detectable by tests [3,4,5].

The rate of bacterial contamination due to the storage conditions of platelets at room temperature—a conducive environment for bacterial proliferation within the product’s shelf life—can lead to life-threatening reactions; 1:10,000 for bacterially contaminated products, and 1:50,000 for septic transfusion reactions. Diversion pouches and better arm disinfection decreased the platelet contamination rate by up to 77% [6], while advanced methodologies like large volume delayed sampling (LVDS) for both aerobic and anaerobic culturing, pathogen reduction technology (PRT), as well as secondary methods such as Verax Biomedical rapid platelet Pan Genera Detection testing have further contributed to decreasing the infection rates. Particularly noteworthy within pathogen reduction technologies (PRT) are techniques relying on ultraviolet illumination in the presence of photoactive substances like amotosalen/psoralen [7] or riboflavin [8], which, despite their proven effectiveness, are not without the potential risks of immune reactions or toxicity [9]. Although these advancements stand as testaments to the evolution of blood safety practices, the risks are not yet negligible.

Current pathogen reduction technologies for blood products include the Mirasol PRT system (TerumoBCT Biotechnologies, Lakewood, CO, USA), the THERAFLEX MB Plasma system (MacoPharma, Tourcoing, France), and the INTERCEPT Blood System (Cerus Corporation, Concord, CA, USA). Mirasol uses riboflavin (vitamin B2) in combination with UVB light (265–370 nm), whereas INTERCEPT relies on amotosalen HCl (a psoralen) and UVA light (320–400 nm). By contrast, THERAFLEX utilizes shortwave UVC irradiation (254 nm) and does not require exogenous photoactive compounds [10]. However, among these, the only FDA-approved PRT method for platelets in the United States is INTERCEPT [11]. Although effective at reducing bacterial contamination, this method is laborious as it includes an adsorption step that removes extraneous amotosalen to prevent a reaction in the eventual patient recipient. Furthermore, with many blood centers having a dual inventory of LVDS and INTERCEPT-PRT, the operational complexity becomes a pain point in platelet production. Therefore, developing a PRT method with a simple workflow and without the need for a small photoactive reagent would be extremely advantageous.

The application of irradiation to biological tissues is a standard sterilization practice for tissue banks. The International Organization for Standardization (ISO) and the American Association of Tissue Banks (AATB) provide guidelines for various tissue types that are routinely irradiation for clinical use. Bone, connective tissue, skin, and vascular grafts commonly receive between 15 and 35 kGy. Corneas and amniotic membranes are treated with between 10 and 25 kGy. These irradiation procedures are critical for reducing the risk of disease transmission, ensuring that patients receive safe and sterile medical tissues.

However, the only example of radiation being employed in transfusion is for transfusion-associated graft versus host disease (TA-GVHD), which is more common in neonatal intrauterine transfusion recipients, certain immunocompromised individuals, recipients of transplants from relatives or HLA-matched donors, and those who have had marrow or stem cell transplants. In this clinical scenario, the cellular target is the donor’s residual white blood cells in the donated product, and the irradiation dose is 25 Gy targeted at the center of the blood product using Food and Drug Administration (FDA)-approved X-ray irradiation cabinets. The radiation dose is delivered at standard clinical dose rates of ~0.1 Gy/s (Quastar™ X-ray emitter), which are low dose rates, making it infeasible to use to reach the antibacterial dose range.

In contrast, linear accelerators such as those used for cancer therapy can be configured to deliver radiation at dose rates several orders of magnitude higher than conventional X-ray irradiation cabinets. In this study, we explore the potential of ultra-high dose rate (UHDR) irradiation to deliver bacterial and viral sterilization dose ranges in blood samples in a matter of minutes. Preclinical studies, both in vivo and in vitro, have utilized UHDR irradiation due to its association with lower toxicity to healthy tissues or cell culture when compared to conventional dose rate (CONV) irradiation. In our study, we utilize UHDR due to the speed at which kGy doses can be delivered. Our hypothesis posits that UHDR irradiation can rapidly sterilize bacteria without adversely affecting the production of clinical-grade apheresis platelet products. We also provide proof-of-concept that linear accelerator-based irradiation can be a practical point-of-care solution for the sterilization of blood products.

We configure a clinical linear accelerator to uniformly administer doses of radiation to multiple 2 mL blood samples at a rate of 100 Gy/s using a 16 MeV electron beam. Both apheresis-derived platelet products and COVID-19 convalescent plasma (CCP) are irradiated with a range of doses up to a maximum of 25 kGy. Our proof-of-concept study demonstrates the capability of UHDR technology as a promising tool for enhancing the security of platelet transfusions and convalescence plasma across different medical scenarios.

## 2. Results

### 2.1. Irradiation and Dosimetry

The beam profiles showed satisfactory flatness—3.4% in the x direction and 4.0% in the y direction—across the 10 × 10 cm^2^ area of the sample holder, which received at least 91.4% of the maximum dose at every point (Figure 1A). Similarly, the dose map also confirmed the relative homogeneity across the sample holder, with a constant dose above 90% of the maximum dose (Figure 1B). The mean dose per pulse measured at the center of the tubes was 0.53 ± 0.02 Gy/pulse (mean ± standard deviation). The mean time of delivery per 1 kGy was 10.44 ± 0.43 s, with a mean of 1887 ± 77 pulses at a 180 Hz pulse rate. The mean dose rate was estimated to be 95.49 ± 3.69 Gy/s (Table 1).

### 2.2. Irradiation Dose–Response on Bacteria Growth

Five platelet product aliquots per dose group from five healthy blood donor samples were spiked with approximately 10^5^ colony-forming units (CFU) per mL of *Escherichia coli* (*E. coli*) prior to irradiation. Samples from each donor were irradiated with a range of doses (0, 0.1, 0.5, 1, 5, 10, and 20 kGy), then plated for growth with CFU enumeration two days later. As shown in Figure 2A, even 1 kGy was able to completely suppress bacterial growth for four out of five spiked samples, with one sample showing a 2-log reduction in bacterial growth. Figure 2B illustrates the average reduction in bacterial growth, which was a 2.7-log reduction from the initial seeding at 1 kGy. By extrapolation from an exponential decay fit, a 6-log bacterial killing dose which is considered to be sterilizing would be 2.3 ± 0.1 kGy.

### 2.3. Irradiation Dose–Response on Platelet Count

We also sought to determine the dose–response of the platelet count over this dose range. We used the same spiked platelet product aliquots from the same donors, and subjected them to 1, 5, 10, and 20 kGy irradiation, including a non-irradiation control. Post-irradiation, we evaluated the platelet count to assess the impact of the different irradiation doses. As shown in Figure 2C and Table 2, the platelet count drops significantly from 1 to 5 kGy. Compared to the platelet count from the unirradiated samples, 1 kGy radiation treatment reduced the platelet count minimally to 95% ± 5% (*p* > 0.05), while 5 kGy reduced the platelet count to 56% ± 11% (** *p* < 0.005) and 20 kGy to 25% ± 3% (*** *p* < 0.001). At the extrapolated 6-log bacterial killing dose of 2.3 ± 0.1 kGy for bacterial sterilization, the platelet count was estimated to be at 69% ± 1% of the unirradiated controls (Figure 2D).

### 2.4. Irradiation Effects on Protein Integrity

Additionally, we sought to determine whether irradiation would have an impact on the protein binding structure in apheresis plasma products. Antibody binding in cognate antigens is highly dependent on an intact protein structure. Given that CCP is known to contain high levels of SARS-CoV-2-specific antibody, we investigated whether sterilizing doses of irradiation can abrogate the ability of IgG-mediated Receptor Binding Domain (RBD) binding in CCP. From ten CCP units of unique donors collected between March 2020 and June 2021, two aliquots of each were obtained for subsequent experimentation. One aliquot from each donor was used for 25 kGy irradiation, a dose chosen to be effective for viral sterilization; the second aliquot was the corresponding non-irradiated control. All samples were thawed and RBD-specific IgG antibody binding was determined by enzyme-linked immunosorbent assay (ELISA), using readout optical density wavelength of 405 nm [12,13]. There was donor-to-donor variability in the magnitude of the binding changes observed with this assay. Specifically for IgG, the range of change was from 2.7% to 19.1%, with a standard deviation of 5.2%. Because no established threshold for efficacy exists for this assay, these results should be interpreted only in terms of relative changes rather than absolute values of antibody binding efficacy. As shown in Figure 3, RBD-specific IgG antibody binding showed a 9.2% drop on average. In contrast to IgG, changes to IgA- and IgM-mediated RBD-binding were not statistically significant (Figure 3).

## 3. Discussion

We have described a novel method to rapidly reduce the bacterial load in apheresis platelet products collected from healthy volunteer blood donors. We could deliver 1 kGy irradiation in ~10 s and 25 kGy in ~3 min. In our pilot study, we have demonstrated that a 1 kGy dose is high enough to significantly reduce bacterial growth from Gram-negative *E. coli* without significantly decreasing the platelet count in the product, and furthermore, that a 25 kGy viral sterilization dose only minimally decreased SARS-CoV-2-specific antibody binding.

Considering bacterial sterilization, we performed bacterial growth dose–response assays using *E. coli* as a pilot organism. Seeding at approximately 10^5^ CFU and with irradiation doses ranging from 0.1 to 20 kGy, we found that even 1 kGy induced complete bacterial growth suppression in four out of five spiked platelet samples at 48 h. We estimated by exponential decay fitting that a sterilizing dose to produce 6-log killing was 2.3 kGy.

Conversely, we have determined the platelet count dose–response to irradiation in the range from 1 to 20 kGy. The platelet count did not significantly decrease for doses of 1 kGy or less. At the estimated 6-log bacterial killing dose of 2.3 kGy, the platelet count was reduced by only 31%, which is encouraging. Transfusion guidelines are typically based on the absolute platelet yield rather than the relative percentage of platelets lost during processing. As these standards can vary across regulatory agencies worldwide, a 31% reduction from baseline may still meet many transfusion thresholds, especially if the initial collection is adjusted to ensure that the final product maintains the minimum acceptable platelet dose.

Moving forward, it is essential to replicate these experiments with a larger sample size and expand them to confirm more precisely that sterilizing doses for a range of pathogens can be administered while maintaining adequate platelet function. Future studies should expand the evaluation to include assays assessing platelet activation and homeostasis. In addition to counting platelets, methods such as light transmission or impedance aggregometry, flow cytometry for markers like P-selectin, and hemostatic tests including thromboelastography (TEG) or rotational thromboelastometry (ROTEM) should be incorporated. Clot retraction and platelet adhesion assays will further elucidate these functional capacities.

To determine the irradiation effects on protein integrity in plasma, we used CCP with a known high titer of SARS-CoV-2-specific antibodies. Since an intact protein structure is crucial for antibody specificity, we assessed whether virus sterilizing doses of irradiation (25 kGy) would decrease RBD-specific antibody binding by ELISA. There was indeed a small but significant drop in IgG binding. However, we will need to determine whether this drop is considered significant in different clinical contexts. Although CCP has been a clinically relevant therapeutic in specific scenarios [14], with vaccination and monoclonals now widely available, as well as changes in COVID-19-related mortality, the small drop in RBD-specific RBD activity may be minimally impactful. However, to further determine the effects of irradiation on protein integrity, it would be beneficial to measure the changes in key proteins in the coagulation cascade related to increasing irradiation doses.

Further studies will also need to confirm sterilizing doses for clinically relevant transfusion-related viruses (e.g., HIV, HCV, HBV, and WNV) as well as emerging infectious diseases for which there are currently no available tests (e.g., dengue and chikungunya viruses). Because viruses generally possess smaller genomes relative to bacteria, the probability of inducing a lethal lesion in each pathogenic organism per unit dose is lower, and the required dose to effectively sterilize viruses in blood products is substantially higher. The literature indicates that 6-log viral inactivation is achieved by 10–50 kGy depending on the virus, medium, and temperature [15,16,17,18,19,20,21]. Moreover, since the mechanism by which irradiation damages cellular material is radical formation, further exploration into the mitigating effects of antioxidants (e.g., N-acetylcysteine) on blood product damage will be crucial, especially if the effects are different between blood components and pathogens.

The current work is presented as a proof-of-concept and requires additional future experiments to confirm and expand on these findings. Using irradiation for pathogen reduction is potentially simpler than current methods involving the addition, activation, and removal of small photo-active compounds. In fact, if eventually engineered appropriately, this usage would be analogous to current low-dose X-ray blood irradiation cabinets: using a small and enclosed chamber to irradiate the platelet product homogeneously before distribution or transfusion. Pathogen reduction can be performed in matter of seconds for bacteria and a few minutes for viruses without product manipulation, instead of hours needed with the currently available PRT devices. Furthermore, it would be straightforward to design dedicated linear accelerator-based irradiators for the practical implementation of this approach at the point of care in blood centers, in contrast to the large, centralized irradiation facilities used for the industrial sterilization of medical products. We are hopeful that this new method of pathogen reduction may prove beneficial to the blood collection industry and transfusion services.

## 4. Materials and Methods

### 4.1. Irradiation

All blood products were irradiated with a clinical linear accelerator (Varian Trilogy, Varian Medical Systems, Palo Alto, CA, USA) configured for UHDR irradiations, as previously described [22,23]. For the irradiation of multiple samples of blood products at the same dose, a 14-slot holder of 2 mL cryovials (internally threaded, Corning™, Corning, NY, USA) was designed in Fusion 360^®^ (Autodesk, San Rafael, CA, USA). The 3D computer-aided design (CAD) files were edited with Ultimaker Cura v.4.3.1 and printed with Ultimaker S5^®^ (New York, NY, USA) using polylactic acid (PLA). The sample holder was designed to allow the vials with the samples to be immersed under liquid and consequently to allow the control of the temperature of the samples during irradiation (Figure 4A–C). The cryovials were irradiated at a source-to-surface distance (SSD; surface of the holder) of 44.1 cm (Figure 4D) with a 20 × 20 cm^2^ jaw opening (actual field size at 100 cm SSD), resulting in an approximately 10 × 10 cm^2^ irradiation field. The beam profiles and dose homogeneity were evaluated using radiochromic film at 44.8 cm SSD (central depth of the tube) with 1 cm solid water build-up (Figure 1A,B). The electron beam irradiation geometry parameters are outlined in Table 1. The bacteria-spiked samples were irradiated at room temperature; however, the samples above 1 kGy were irradiated with sample containers immersed in pre-chilled water to avoid a rise in temperature from radiation. The CCP samples were irradiated in a frozen state and immersed in dry-ice-cold isopentane (2′-Methybutane, Sigma-Aldrich, St. Louis, MO, USA) containing pieces of dry ice to maintain a consistent temperature of approximately −78.5 °C throughout the duration of irradiation. At the conclusion of irradiation, the samples were returned to dry ice, before being returned to −80 °C storage prior to further processing.

### 4.2. Dosimetry

The beam was monitored during irradiation by measuring the Bremsstrahlung tail of the electron beam using an ionization chamber (Farmer^®^ 30010, PTW-Freiburg, Freiburg, Germany) placed at 59 cm SSD, upstream of the sample holder and in 13 cm solid water. The measurements of charge were corrected for temperature and pressure. In order to calibrate the dose-to-blood at the depth of the center of the tube oriented perpendicular to the beam (Figure 1B), 2.4 × 5.1 cm^2^ pieces of radiochromic film (Gafchromic EBT-XD, Ashland™, Wayne, NJ, USA) were placed inside an empty holder filled with water (44.8 cm SSD). The films were analyzed as described previously [24]. A fixed number of pulses was delivered per film and, consequently, charges from the ion chamber measurements were correlated to the associated absorbed dose from each radiochromic film, providing a mean measured charge per dose (nC/Gy). The machine output was calibrated on a daily basis prior to each experiment. The electron beam irradiation experimental parameters are outlined in Table 1. To generate the beam profiles and dose maps, a 15 × 15 cm^2^ radiochromic film was inserted into an empty, water-filled holder (44.8 cm SSD). The profiles in both the x and y directions were then evaluated for flatness by comparing the maximum and minimum doses measured within 5 cm of each side of the isocenter, and presented as a percentage (Equation (1)).(1)Flatness=100×((Max−Min))/((Max+Min))

### 4.3. Bacteria Culture and Preparation

*Escherichia coli* (*E. coli* ATCC™ 25922™), obtained as a Culti-Loops™ culture (Thermo Scientific, Waltham, MA, USA), was revived in 500 μL tryptone soy broth at 37 °C for 5–10 min. The resulting bacterial suspension was streaked in the usual fashion onto blood agar plates and incubated at 37 °C for 48 h.

### 4.4. Platelet Components

Transfusable platelet component products were collected from volunteer blood donors (n = 5) at the Stanford Blood Center according to the Association for the Advancement of Blood and Biotherapies (AABB) and Food and Drug Administration (FDA) guidelines [25]. All platelet products were collected by automation. As part of the normal quality check (QC) process, after collection, the platelet products were transported to the lab where a platelet count was performed on each bag by the Sysmex XN2100 instrument (Sysmex, Lincolnshire, IL, USA). A small sample of these platelet products, with the platelet count, was aliquoted into 2 mL cryovials (internally threaded, Corning™, Corning, NY, USA) that was to be sent for irradiation. One aliquot of 2 mL was allocated for each irradiation group: 0 (non-irradiated control), 0.1, 0.5, 1, 5, 10, and 20 kGy. All aliquots were also spiked with 10^5^ CFU of *E. coli* and stored at room temperature on a shaking incubator until irradiation. The bacterial stock was received from the clinical microbiology lab. The stock was titrated and standardized using limiting dilutions before being divided into equal frozen aliquots for future experimentation. All experiments and results presented herein were generated from the same stock that was equally aliquoted and therefore should have the same starting CFUs. After irradiation, platelet counts were performed for each sample, and the samples were immediately cultured on blood agar plates using a sterile calibrated loop at 37 °C for 24 h for colony counting.

### 4.5. Plasma Component

COVID-19 convalescent plasma (CCP) products were also collected by automated method from volunteer blood donors at the Stanford Blood Center according to the overall blood donor collection regulations, and specifically according to FDA Guidance for the recruitment and collection of CCP [26]. At the time of collection, from each donor (n = 10), additional serum tubes were collected, aliquoted into 2 mL cryovials, and stored at −80 °C until needed for further experiments. For each of the ten CCP donors, one aliquot was sent for irradiation in a frozen state and the other was kept as a frozen non-irradiated control. After irradiation, both sets were thawed and SARS-CoV-2 antibody assays were performed to quantify any changes in binding activity.

### 4.6. SARS-CoV-2 Antibody Binding Assay

Previously described in detail [12], and briefly described here, we measured the antibody binding activity in the serum collected from healthy donors known to previously have been diagnosed with COVID-19. We employed an enzyme-linked immunosorbent assay (ELISA)-based assay against Wuhan-Hu-1 SARS-CoV-2 (WT) receptor binding domain (RBD) of the Spike protein as the target. The 96-well plates were coated with 0.1 µg per well of WT-RBD (ATUM) in PBS and incubated overnight at 4 °C, then blocked with PBS-T containing 3% milk. CCP samples were diluted 1:1000, transferred to plates, and incubated at 37 °C for one hour. Horseradish peroxidase-conjugated goat anti-human immunoglobulin G (IgG), IgA, or IgM were used to detect isotype-specific binding. A total of 100 µL of 3,3′,5,5′-Tetramethylbenzidine substrate solution was added to each well, which developed for 12 min before 100 µL of 0.16 M sulfuric acid was added to stop the reaction. The optical density (OD) at 450 nm was measured with a SpectraMax M2 microplate reader (Molecular Devices, San Jose, CA, USA).

### 4.7. Statistics

For the dosimetry, the values presented are the averages of the measurements from all dose groups across multiple independent experiments (n = 5). For the analysis of the colony forming unit and platelet count assays, the dose groups (n = 5 donors per group) were compared to the non-irradiated controls using paired t-tests. A simple exponential decay fit was used to extrapolate the dose required for a 6-log bacterial killing. The percentage of the remaining platelet count was estimated using a one-phase decay (exponential decay with plateau) fit. For the isotype-specific binding assays, each sample (n = 10 donors) was measured in duplicate. All values are presented as mean ± standard deviation.

## 5. Conclusions

This study provides a proof-of-concept for the rapid sterilization of clinical blood products using UHDR irradiation with a clinical linear accelerator. Our results indicate that irradiation of 5 kGy will complete suppression. The estimated 6-log bacterial reduction dose (2.3 kGy) should maintain approximately 70% of the platelet count. Additionally, the viral sterilization of COVID-19 convalescent plasma at 25 kGy resulted in a minimal reduction in RBD-specific IgG binding, suggesting that functional antibody activity is largely preserved. These findings indicate that UHDR irradiation could serve as a promising alternative to existing pathogen reduction technologies by offering rapid and effective blood sterilization without the need for chemical additives. Future studies should expand on these findings to assess the functional viability of irradiated platelets and further evaluate viral inactivation across a broader range of transfusion-relevant pathogens. If validated, this approach could facilitate a streamlined, point-of-care blood sterilization method, enhancing transfusion safety and reducing the risks associated with bacterial and viral contamination in blood products.

## Figures and Tables

**Figure 1 ijms-26-02424-f001:**
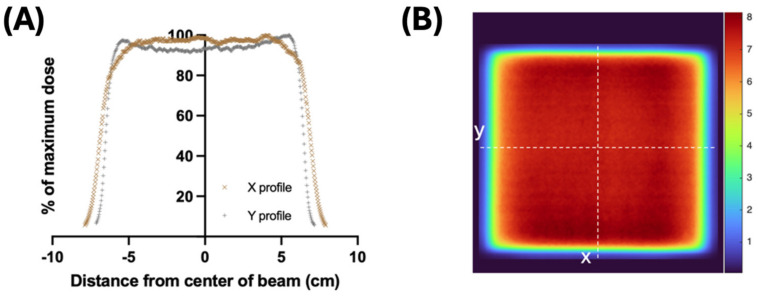
(**A**) The profiles of the beam are derived using film irradiated with 1 cm build-up at 44.8 cm from the source (central depth of the tube) and are relatively flat and comparable in both the x and y directions. (**B**) Dose map of the same films demonstrate relative homogeneity across the 10 × 10 cm^2^ area of interest, with 3.4% and 4.0% flatness in the x and y direction, respectively.

**Figure 2 ijms-26-02424-f002:**
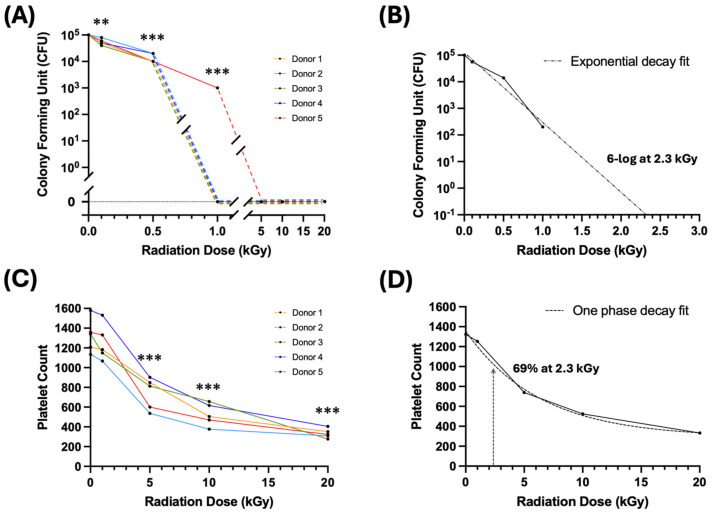
Dose–response of *Escherichia coli* (*E. coli*) bacterial survival (**A**,**B**) and platelet count (**C**,**D**) from irradiated apheresis platelet products acquired from five healthy volunteer donors (one aliquot per dose per donor), each spiked with 10^5^ colony-forming units (CFU) of *E. coli*. (**A**,**B**) Bacterial survival in spiked platelet products post-irradiation per individual donor (**A**), and averaged over all donors (**B**) (dotted line is exponential decay fit). At a 1 kGy dose, only one sample (1 donor) showed a 2-log bacterial reduction, with no growth in the rest, while at 5, 10, and 20 kGy there was no bacterial growth in any sample. The extrapolated sterilizing 6-log killing dose is 2.3 ± 0.1 kGy. (**C**,**D**) Platelet counts in the same spiked platelet products post-irradiation per individual donor (**C**), and averaged over all donors (**D**) (dotted curve is one phase decay [exponential decay with plateau] fit). At a 1 kGy dose, the platelet count reduced minimally to 95% ± 5%. At an estimated 6-log killing dose of 2.3 kGy, the platelet count is estimated to be 69% ± 1%. Paired t-tests; ** *p* < 0.005; *** *p* < 0.001; all comparisons are against a non-irradiated control.

**Figure 3 ijms-26-02424-f003:**
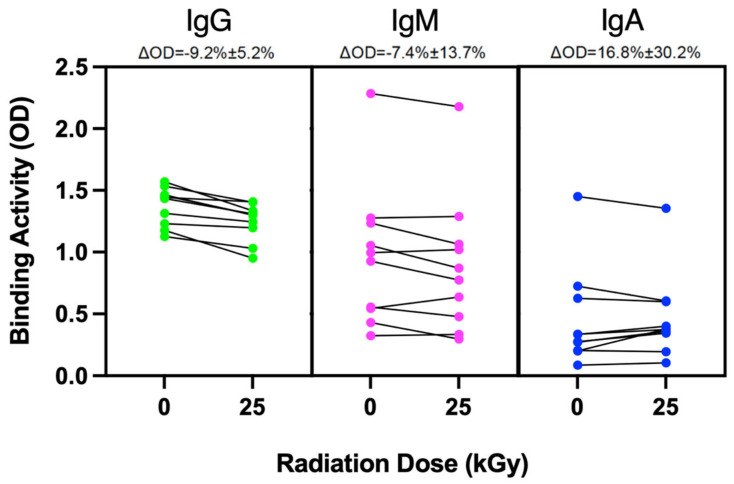
Effects of virus sterilizing radiation dose on SARS-CoV-2 RBD-specific antibodies in COVID-19 convalescent plasma (CCP). A total of 10 CCP donor samples were aliquoted for non-irradiated control and high-dose (25 kGy) irradiation. After irradiation, the level of isotype-specific antibody binding was measured by enzyme-linked immunosorbent assay (ELISA), with readout optical density wavelength of 405 nm. Each sample aliquot was tested in duplicate and the values represent the mean ± standard deviation for IgG, IgM, and IgA RBD-binding.

**Figure 4 ijms-26-02424-f004:**
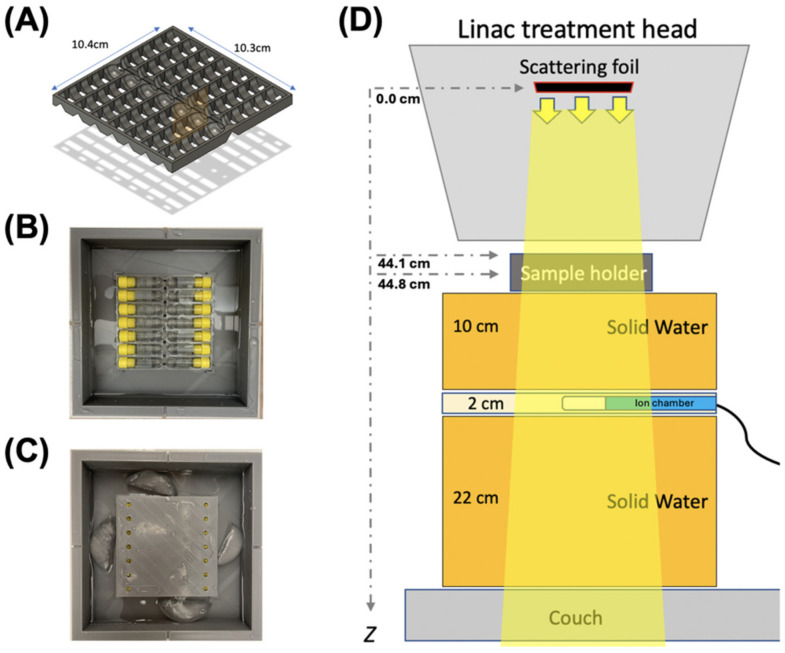
(**A**–**C**) Temperature controlled 14-slot blood sample holder for UHDR irradiation and (**D**) beam geometry. (**A**) CAD illustrates a 14-slot holder for a 2 mL cryovial that allows samples to be immersed in liquid (**B**), and which can be used to control temperature. (**C**) Samples at ~4 °C. (**D**) Schematic diagram illustrates the geometry of the irradiation, with samples at 44.1 cm source-to-surface distance (SSD; surface of the holder from the scattering foil). The beam is monitored using an ion chamber (Farmer chamber) that measures the Bremsstrahlung tail of the electron beam.

**Table 1 ijms-26-02424-t001:** Electron beam irradiation geometry values and experimental parameters.

Parameter	Value
Beam energy [MeV]	16.6
Source-to-surface distance (cm)	44.8
Field size (cm^2^)	10 × 10
Target dose [kGy]	0.1, 0.5, 1, 5, 10, 15, 20, 25
Pulse rate [Hz]	180
Mean dose per pulse [Gy]	0.53
Mean dose rate [Gy/s]	95.5
Pulse length [s]	3.75 × 10^−6^
Mean delivery time per kGy [s]	10.44
Intra-pulse dose rate [Gy/s]	1.41 × 10^5^

**Table 2 ijms-26-02424-t002:** Dose–response of platelet count as percent of non-irradiated control.

Irradiation (kGy)	PLT (A)	PLT (B)	PLT (C)	PLT (D)	PLT (E)	*p*-Value (Compared to 0 kGy)
0	100%	100%	100%	100%	100%	
1	98%	94%	86%	97%	98%	0.0784
5	70%	47%	61%	57%	44%	0.0007
10	42%	33%	49%	39%	35%	0.0000
20	29%	27%	21%	26%	24%	0.0000

## Data Availability

The data sets used and/or analyzed during the current study are available from the corresponding author on reasonable request.

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
