# Peer review of "Rapid Sterilization of Clinical Apheresis Blood Products Using Ultra-High Dose Rate Radiation"

_ijms, 2025, doi:10.3390/ijms26062424_

Round 1
Reviewer 1 Report
Comments and Suggestions for Authors
Melemenidis et al reported an interesting experience on irradiation of blood components (plasma and platelets) to prevent microbial contamination. The idea is very interesting and intriguing. I suggest to the authors to add ,in the discussion some preliminary data about the costs of the procedure, especially compared to methods already present such as in house pathogen inactivation.
Author Response
We thank the reviewer for the thoughtful feedback and careful consideration. Your suggestions have been helpful in improving this work.
Reviewer 1:
Melemenidis et al reported an interesting experience on irradiation of blood components (plasma and platelets) to prevent microbial contamination. The idea is very interesting and intriguing. I suggest to the authors to add ,in the discussion some preliminary data about the costs of the procedure, especially compared to methods already present such as in house pathogen inactivation.
We agree that this would be a great approach to demonstrate the practicality of high-dose irradiation sterilization of platelets. However, since manufacturers have different contracts with each client, the material cost to the blood collection center is considered proprietary and we cannot disclose it.

Reviewer 2 Report
Comments and Suggestions for Authors
General comments:
Pathogen inactivation of blood products is a very relevant topic and simplification of currently used methods is of great importance. Therefore, the article should be published after some improvements. In the manuscript the pathogen inactivation method “Intercept” is mentioned. This method is currently the most widely used method worldwide. In addition, the company Macopharma has developed another method called “Theraflex”, which also works with pure irradiation without a photoactive substance. This method should be mentioned in the discussion and compared with the method described.
Minor Comments:
Introduction:
- Lines 100 to 108: Results are discussed that do not belong in the introduction
Results:
- In the description of Figure 1, Figure (B) is incorrectly labeled as Figure (F)
- The statistical naming of the average values is missing. Is it mean ± standard deviation? The naming and standard deviation is also missing in table 1.
- In the description of Figure 2, (A,B) and (C-D) is not named uniformly.
- Line 165: The abbreviation RBD was not explained.
- It is not clear how the frozen second aliquot of the CCP serves as a non-irradiated control. Please comment on that.
Discussion:
- Line 189: 105 instead of 10⁵
Material and Methods:
- Line 332: There are two dots at the end of the sentence.
Author Response
We thank the reviewer for the thoughtful feedback and careful consideration. Your suggestions have been helpful in improving this work.
Reviewer 2:
General comments:
Pathogen inactivation of blood products is a very relevant topic and simplification of currently used methods is of great importance. Therefore, the article should be published after some improvements. In the manuscript the pathogen inactivation method “Intercept” is mentioned. This method is currently the most widely used method worldwide. In addition, the company Macopharma has developed another method called “Theraflex”, which also works with pure irradiation without a photoactive substance. This method should be mentioned in the discussion and compared with the method described.
We thank the reviewer for the addition. We have updated the Introduction, as follows.
Line 61-69: Current pathogen reduction technologies for blood products include the Mirasol PRT system (TerumoBCT Biotechnologies, Lakewood, CO, USA), the THERAFLEX MB Plasma system (MacoPharma, Tourcoing, France), and the INTERCEPT Blood System (Cerus Corporation, Concord, CA, USA). Mirasol uses riboflavin (vitamin B2) in combination with UVB light (265–370 nm), whereas INTERCEPT relies on amotosalen-HCl (a psoralen) and UVA light (320–400 nm). By contrast, THERAFLEX utilizes shortwave UVC irradi-ation (254 nm) and does not require exogenous photoactive compounds [10]. However, among these, the only FDA-approved PRT method for platelets in the United States is INTERCEPT [11].
- Picker, S.M. Current Methods for the Reduction of Blood-Borne Pathogens: A Comprehensive Literature Review. Blood Transfus 2013, 11, 343–348, doi:10.2450/2013.0218-12.
- Investigational COVID-19 Convalescent Plasma; Guidance for Industry.
Minor Comments:
Introduction:
- Lines 100 to 108: Results are discussed that do not belong in the introduction
We agree with the reviewer, and we have removed the results from the Introduction section. The last paragraph of the Introduction now reads:
Line 104-110: We have configured a clinical linear accelerator to uniformly administer doses of radiation to multiple 2 mL blood samples at a rate of 100 Gy/s using a 16 MeV electron beam. Both apheresis-derived platelet products and COVID-19 convalescent plasma (CCP) were irradiated with a range of doses up to a maximum of 25 kGy. Our proof-of-concept study demonstrates the capability of UHDR technology as a promising tool for enhancing the security of platelet transfusions and convalescence plasma across different medical scenarios.
Results:
- In the description of Figure 1, Figure (B) is incorrectly labeled as Figure (F)
Corrected
- The statistical naming of the .values is missing. Is it mean ± standard deviation? The naming and standard deviation is also missing in table 1.
In the manuscript, Methods section, 4.7 Statistics, we state:
Line 331-332, of previously submitted manuscript: All values are presented as mean ± SD, except for the isotype-specific binding values, which are presented as the average of duplicates ± SD.
We have made the following editorial changes to the manuscript to enhance clarity and readability.
Lines: 113-120: 2.1. Irradiation and dosimetry
The beam profiles showed satisfactory flatness—3.4% in the x-direction and 4.0% in the y-direction—across the 10 x 10 cm² area of the sample holder, which received at least 91.4% of the maximum dose at every point (Figure 1A). Similarly, the dose map also demonstrated homogeneity across the sample holder with dose consistently above 90% of the maximum dose (Figure 1B). The mean dose per pulse measured at the center of the tubes was 0.53 ± 0.02 Gy/pulse (mean ± standard deviation). The mean time of delivery per 1 kGy was 10.44 ± 0.43 seconds, with a mean of 1887 ± 77 pulses at 180 Hz pulse rate. The mean dose rate was estimated to be 95.49 ± 3.69 Gy/s (Table 1).
Line 127: Table 1. Electron Beam Irradiation Geometry Values and Experimental Parameters.
Parameter |
Value |
Beam energy [MeV] |
16.6 |
Source to surface distance (cm) |
44.8 |
Field size (cm2) |
10×10 |
Target dose [kGy] |
0.1, 0.5, 1. 5, 10, 15, 20, 25 |
Pulse rate [Hz] |
180 |
Mean dose per pulse [Gy] |
0.53 |
Mean dose rate [Gy/s] |
95.5 |
Pulse length [s] |
3.75E-6 |
Mean delivery time per kGy [s] |
10.44 |
Intra-pulse dose rate [Gy/s] |
1.41E+5 |
Figure 3 legend now reads:
Lines 182-186: Figure 3. Effects of virus sterilizing radiation dose on SARS-CoV-2 RBD-specific antibodies in COVID Convalescent Plasma (CCP). 10 CCP donor samples were aliquoted for non-irradiated control and high-dose (25 kGy) irradiation. After irradiation, level of isotype-specific antibody binding was measured by ELISA, with OD405 readout. Each sample aliquot was tested in duplicates and values represent as mean ± standard deviation for IgG, IgM, and IgA RBD-binding.
Line 356-359: The percentage of remaining platelet count was estimated using a one-phase decay (exponential decay with plateau) fit. For the isotype-specific binding assays, each sample (n = 10 donors) was measured in duplicates. All values are presented as mean ± standard deviation.
- In the description of Figure 2, (A,B) and (C-D) is not named uniformly.
We have attempted to clarify the caption for Figure 2.
Figure 2 legends now reads:
Line 139-150: Dose-response of E.coli bacterial survival (A)&(B) and platelet count (C)&(D) from irradiated apheresis platelet products acquired from five healthy volunteer donors (one aliquot per dose per donor) each spiked with 105 CFU of E.coli. (A)&(B) Bacterial survival in spiked platelet products post-irradiation per individual donor (A), and averaged over all donors (B) (dotted line is expo-nential decay fit). At 1 kGy dose only one sample (1 donor) showed 2-log bacterial reduction with no growth in the rest, while at 5, 10 and 20 kGy there was no bacterial growth in any sample. The extrapolated sterilizing 6-log killing dose is 2.3 ± 0.1 kGy. (C)&(D) Platelet counts in the same spiked platelet products post-irradiation per individual donor (C), and averaged over all donors (D) (dotted curve is one phase decay [exponential decay with plateau] fit). At 1 kGy dose, platelet count reduced minimally to 95% ± 5%. At estimated 6-log killing dose of 2.3 kGy the platelet count is estimated to be 69% ± 1%. Paired t-tests; *P < 0.01; **P < 0.005; ***P < 0.001; all comparisons are against non-irradiated control.
We also changed Figure 2, to correct (B) for saying ‘Exponential growth fit’, instead of Exponential decay fit.
- Line 165: The abbreviation RBD was not explained.
‘Receptor Binding Domain’ explanation was added before the first abbreviation in the manuscript.
- It is not clear how the frozen second aliquot of the CCP serves as a non-irradiated control. Please comment on that.
In Methods section, 4.5 Plasma component, we state:
Line 306-309, of previously submitted manuscript: For each of ten CCP donors, one aliquot was sent for irradiation in frozen state and the other was kept as frozen non-irradiated control. After irradiation, both sets were thawed and SARS-CoV2 antibody assays were performed to quantify any changes in binding activity.
Discussion:
- Line 189: 105 instead of 10⁵
Corrected
Material and Methods:
- Line 332: There are two dots at the end of the sentence.
Corrected

Reviewer 3 Report
Comments and Suggestions for Authors
The study explores rapid irradiation as an innovative pathogen reduction method that maintains platelet counts and CCP antibody binding at sterilizing doses, highlighting its potential as a point-of-care blood product sterilization solution.
I recommend to accept this manuscript, but after a Major Revision, because I have a few concerns that must be addressed first:
- Inconsistencies with sample temperature. Platelet samples were irradiated at room temperature (above 1kGy in pre-chilled water), while CCP plasma was irradiated frozen (-78.5°C). Temperature differences may affect radiation damage mechanisms (e.g. free radical formation). No discussion on how such conditions may affect the comparability of results.
- No data regarding dose homogeneity. Despite the claimed "flatness" of the beam (Fig. 1A-B), no precise data on dose variability within samples were provided. Even slightly non-uniform irradiation could have distorted the results, especially at high doses (>10 kGy).
- No information on standardization of initial bacterial concentration (105 CFU/mL) during E. coli inoculation procedure. Differences in initial contamination may have contributed to sample-to-sample variability.
- The estimated dose of 2.3 kGy (Fig. 2B) is based on an exponential model, but the data for 1 kGy show a nonlinear decline (4/5 samples with no increase). The model does not consider a possible "threshold" of sterilization efficacy, leading to an underestimation of the dose.
- In section 2.3. Authors demonstrate decrease in platelet count. At 1 kGy: 95% platelet count (no statistical significance). At 2.3 kGy (extrapolation): 69% platelet retention. Actual data at 5 kGy indicates a decrease of up to 56%. The single-phase decay model (Fig. 2D) may not reflect the actual damage kinetics.
- At 2.3 kGy, 69% platelets remain, but there is no discussion on whether this number is sufficient for transfusion.
- Platelet count does not equal to platelet functionality. Without data on activation/hemostasis capacity, results are incomplete.
- In section 2.4. Authors demonstrate decrease in IgG binding. Average decrease of 9.2% at 25 kGy (Fig. 3), but no data on interdonor variability. Is the decrease clinically significant? No reference to a minimum threshold for antibody efficacy.
- The reduction in IgG binding is explained by "structural damage", but there is no analysis of e.g. thermal denaturation or the influence of radicals.
- In plasma sterilization studies, typical doses are 15-35 kGy (mentioned in the Introduction), but the authors do not refer to these values when interpreting the results for 25 kGy.
Author Response
We thank the reviewer for the thoughtful feedback and careful consideration. Your suggestions have been helpful in improving this work.
Reviewer 3:
I recommend to accept this manuscript, but after a Major Revision, because I have a few concerns that must be addressed first:
1. Inconsistencies with sample temperature. Platelet samples were irradiated at room temperature (above 1kGy in pre-chilled water), while CCP plasma was irradiated frozen (-78.5°C). Temperature differences may affect radiation damage mechanisms (e.g. free radical formation). No discussion on how such conditions may affect the comparability of results.
The sample temperature was consistent within each sample type. The choice of temperature for each sample type was chosen to match clinical conditions; platelets are maintained at room temperature and plasma is frozen. The reviewer is right to point out that the ‘temperature differences may affect radiation damage mechanisms’. However, for each sample type, the irradiation was performed at a consistent temperature, and comparisons are made only between samples of the same type, and not across sample types.
2. No data regarding dose homogeneity. Despite the claimed "flatness" of the beam (Fig. 1A-B), no precise data on dose variability within samples were provided. Even slightly non-uniform irradiation could have distorted the results, especially at high doses (>10 kGy).
In order to clarify the flatness, we had added calculations of the flatness in both x and y directions and made the appropriate changes in the manuscript. ‘Results’ section 2.1 now reads:
Line 113-115: The beam profiles showed satisfactory flatness—3.4% in the x-direction and 4.0% in the y-direction—across the 10 × 10 cm² area of the sample holder, which received at least 91.4% of the maximum dose at every point (Figure 1A).
Figure 1 caption now reads:
Line 122-125: (A) The profiles of the beam were derived using film irradiated with 1 cm build up at 44.8 cm from the source (central depth of the tube) and are relatively flat and comparable in both x and y directions. (B) Dose map of the same films demonstrate relative homogeneity across the 10 × 10 cm2 area or interest, with 3.4% and 4.0% flatness in x and y direction respectively.
‘Methods’ section 4.2 Dosimetry now reads:
Line 298-304: To generate beam profiles and dose maps, a 15 × 15 cm² radiochromic film was inserted into an empty, water-filled holder (44.8 cm SSD). The profiles in both the x and y directions were then evaluated for flatness by comparing the maximum and minimum doses measured within 5 cm of each side of the isocenter, and presented as a percentage (Equation 1).
3. No information on standardization of initial bacterial concentration (105 CFU/mL) during E. coli inoculation procedure. Differences in initial contamination may have contributed to sample-to-sample variability.
We have added additional information to the Methods section providing standardization information. 4.4 Platelet components section now reads:
Line 319-327: All aliquots were also spiked with 105 CFU of E. coli and stored at room temperature on a shaking incubator until irradiation. Bacterial stock was received from clinical microbiology lab. The stock was titrated and standardized using limiting dilutions before being divided into equal frozen aliquots for future experimentation. All experiments and results presented herein were generated from the same stock that was equally aliquoted and therefore should have the same starting CFUs. After irradiation, platelet counts were performed for each sample, and samples were immediately cultured on blood agar plates using a sterile calibrated loop at 37°C for 24 hours for colony counting.
4. The estimated dose of 2.3 kGy (Fig. 2B) is based on an exponential model, but the data for 1 kGy show a nonlinear decline (4/5 samples with no increase). The model does not consider a possible "threshold" of sterilization efficacy, leading to an underestimation of the dose.
This is a proof-of-concept study demonstrating at a high level that a pathogen sterilizing dose of radiation can be compatible with platelet survival. This finding does not depend on a specific model. We found that at 1 kGy, 80% of samples had no bacterial growth, and at 5 kGy and above, no samples had any bacterial growth. The dose to achieve sterilization is therefore between 1 and 5 kGy. The exponential decay in Fig. 2B was a simple model that provided a reasonable fit to our data and yielded an estimate of 2.3 kGy for 6-log killing. Other models could be used, but would add complexity beyond the basic scope of this pilot work. Of note, modeling a threshold dose with our data would result in a steeper slope after the threshold and a lower estimated 6-log killing dose, making the estimate we used a conservative one.
5. In section 2.3. Authors demonstrate decrease in platelet count. At 1 kGy: 95% platelet count (no statistical significance). At 2.3 kGy (extrapolation): 69% platelet retention. Actual data at 5 kGy indicates a decrease of up to 56%. The single-phase decay model (Fig. 2D) may not reflect the actual damage kinetics.
We appreciate the reviewer’s feedback. Again, a one-phase decay model provided a simple and reasonable fit to our data, but our conclusions do not depend on this specific model. It merely provides a way of interpolating platelet counts between our measured data points at 1 and 5 kGy. As can be seen from the plot, any other fitting model would produce similar values.
6. At 2.3 kGy, 69% platelets remain, but there is no discussion on whether this number is sufficient for transfusion.
We appreciate the reviewer’s comment. The current standards among different countries are based on the total number of platelets in the unit at the time of release and not based on percentage survival. Furthermore, these standards for the minimum platelet yield per unit varies among different countries, as regulatory agencies of each country determine what is ideal. Additionally, if there should be a higher percentage drop in platelet count, this can be anticipated and mitigated by targeting a higher platelet yield at the time of collection.
We have added this information to the Discussion:
Line 203-208: At the estimated 6-log bacterial killing dose of 2.3 kGy, the platelet count is reduced by only 31%, which is encouraging. Transfusion guidelines are typically based on the absolute platelet yield rather than the relative percentage of platelets lost during processing. As these standards can vary across regulatory agencies worldwide, a 31% reduction from baseline may still meet many transfusion thresholds, especially if the initial collection is adjusted to ensure the final product maintains the minimum acceptable platelet dose.
7. Platelet count does not equal to platelet functionality. Without data on activation/hemostasis capacity, results are incomplete.
We agree with the reviewer that platelet count does not equate with platelet functionality. In our work we do not claim that the two are one and the same and at Discussion section we specifically state:
Line 197-199, of previously submitted manuscript: Moving forward, it is essential to replicate these experiments with a larger sample size and expand them to confirm more precisely that sterilizing doses for a range of pathogens can be administered while maintaining adequate platelet function.
For further completion we have added to this paragraph:
Line 211-217: Future studies should expand the evaluation to include assays assessing platelet activation and homeostasis. In addition to counting platelets, methods such as light transmission or impedance aggregometry, flow cytometry for markers like P-selectin, and hemostatic tests including thromboelastography (TEG) or rotational thromboelastometry (ROTEM) should be incorporated. Clot retraction and platelet adhesion assays will further elucidate these functional capacities.
8. In section 2.4. Authors demonstrate decrease in IgG binding. Average decrease of 9.2% at 25 kGy (Fig. 3), but no data on interdonor variability. Is the decrease clinically significant? No reference to a minimum threshold for antibody efficacy.
There is donor variability in the change in binding. For IgG, the range was from 2.7% to 19.1%, with a standard deviation of 5.2%. For the assay that we used, there was no established threshold for efficacy and therefore should only be interpreted in the context of relative change.
We have changed the results to include this information for clarity. The Results section, 2.4. Irradiation effects on protein integrity now reads.
Line 173-180: All samples were thawed and RBD-specific IgG antibody binding was determined by ELISA [12,13]. There is donor-to-donor variability in the magnitude of binding changes observed with this assay. Specifically for IgG, the range of change was from 2.7% to 19.1%, with a standard deviation of 5.2%. Because no established threshold for efficacy exists for this assay, these results should be interpreted only in terms of relative changes rather than absolute values of antibody-binding efficacy. As shown in Figure 3, RBD-specific IgG antibody binding showed a 9.2% drop on average. In contrast to IgG, changes to IgA and IgM mediated RBD-binding were not statistically significant (Figure 3).
9. The reduction in IgG binding is explained by "structural damage", but there is no analysis of e.g. thermal denaturation or the influence of radicals.
It has been shown that high-dose irradiation can indeed alter protein structure in IgG and that it can be mitigated (Smeltzer et al, 2015). Therefore, we speculated that it could be due to structural change, but to prove this would be beyond the scope of this proof-of-concept manuscript.
Smeltzer CC, Lukinova NI, Towcimak ND, Yan X, Mann DM, Drohan WN, Griko YV. Effect of gamma irradiation on the structural stability and functional activity of plasma-derived IgG. Biologicals. 2015 Jul;43(4):242-9. doi: 10.1016/j.biologicals.2015.04.003. Epub 2015 May 8. PMID: 25962339.
10. In plasma sterilization studies, typical doses are 15-35 kGy (mentioned in the Introduction), but the authors do not refer to these values when interpreting the results for 25 kGy.
We appreciate this observation. In the Introduction, we cite the ISO and AATB guidelines, which typically recommend 15–35 kGy for sterilizing various tissue products, not plasma. Our choice of 25 kGy for plasma sterilization falls within this range and aligns with published studies showing that 6-log viral inactivation can be achieved at doses of roughly 10–50 kGy, depending on the virus, medium, and temperature as referenced in the Discussion section of the submitted manuscript:
Lines 217-218: Literature indicates that for 6-log viral inactivation 25 kGy appears to be on the upper end of the range for irradiation conditions above 0°C [14–20].
- House, C.; House, J.A.; Yedloutschnig, R.J. Inactivation of Viral Agents in Bovine Serum by Gamma Irradiation. Can J Micro-biol 1990, 36, 737–740, doi:10.1139/m90-126.
- Hume, A.J.; Ames, J.; Rennick, L.J.; Duprex, W.P.; Marzi, A.; Tonkiss, J.; Mühlberger, E. Inactivation of RNA Viruses by Gam-ma Irradiation: A Study on Mitigating Factors. Viruses 2016, 8, 204, doi:10.3390/v8070204.
- Sullivan, R.; Fassolitis, A.C.; Larkin, E.P.; Read, R.B.; Peeler, J.T. Inactivation of Thirty Viruses by Gamma Radiation. Appl Microbiol 1971, 22, 61–65, doi:10.1128/am.22.1.61-65.1971.
- Kitchen, A.D.; Mann, G.F.; Harrison, J.F.; Zuckerman, A.J. Effect of Gamma Irradiation on the Human Immunodeficiency Vi-rus and Human Coagulation Proteins. Vox Sang 1989, 56, 223–229, doi:10.1111/j.1423-0410.1989.tb02033.x.
- Hiemstra, H.; Tersmette, M.; Vos, A.H.; Over, J.; van Berkel, M.P.; de Bree, H. Inactivation of Human Immunodeficiency Virus by Gamma Radiation and Its Effect on Plasma and Coagulation Factors. Transfusion 1991, 31, 32–39, doi:10.1046/j.1537-2995.1991.31191096182.x.
- Grieb, T.; Forng, R.-Y.; Brown, R.; Owolabi, T.; Maddox, E.; McBain, A.; Drohan, W.N.; Mann, D.M.; Burgess, W.H. Effective Use of Gamma Irradiation for Pathogen Inactivation of Monoclonal Antibody Preparations. Biologicals 2002, 30, 207–216, doi:10.1006/biol.2002.0330.
- Feldmann, F.; Shupert, W.L.; Haddock, E.; Twardoski, B.; Feldmann, H. Gamma Irradiation as an Effective Method for Inac-tivation of Emerging Viral Pathogens. Am J Trop Med Hyg 2019, 100, 1275–1277, doi:10.4269/ajtmh.18-0937.
Because our work is a proof-of-concept study, we selected 25 kGy as a representative sterilization dose. However, we agree that a more comprehensive dose-response evaluation—encompassing a range of pathogens and blood product types—will be needed to refine these parameters further, as we mention in the Discussion section.
For additional clarity, we have revised the manuscript to explicitly indicate the broader range of doses reported in the literature for viral inactivation. The relevant portion of the Discussion now reads:
Lines 235-236: Literature indicates that for 6-log viral inactivation is achieved by 10–50 kGy depending on the virus, medium, and temperature [15–21].

Round 2
Reviewer 3 Report
Comments and Suggestions for Authors
Authors improved the manuscript significantly in response to the previous review comments and now in my opinion the manuscript is ready for publication.